# Position: Towards Responsible Evaluation for Text-to-Speech

Yifan Yang [* 1]   Hui Wang [* 2]   Bing Han [1]   Shujie Liu [3]   Jinyu Li [3]   Yong Qin [2]   Xie Chen [1 4]

## Abstract

Recent advances in text-to-speech (TTS) technology have enabled systems to generate speech that is often indistinguishable from human speech, bringing benefits to accessibility, content creation, and human-computer interaction. However, current evaluation practices are increasingly inadequate for capturing the full range of capabilities, limitations, and societal impacts of modern TTS systems. This position paper introduces the concept of *Responsible Evaluation* and argues that it is essential and urgent for the next phase of TTS development, structured through three progressive levels: (1) ensuring the faithful and accurate reflection of a model's true capabilities and limitations, with more robust, discriminative, and comprehensive objective and subjective scoring methodologies; (2) enabling comparability, standardization, and transferability through standardized benchmarks, transparent reporting, and transferable evaluation metrics; and (3) assessing governance, fairness, and security concerns around data provenance, disparities, misuse, spoofing, and traceability. Through this concept, we critically examine current evaluation practices, identify systemic shortcomings, and propose actionable recommendations. We hope this concept will not only foster more reliable TTS technology but also guide its development toward ethically sound and societally beneficial applications.

## 1. Introduction

Text-to-speech (TTS) has advanced rapidly in recent years, driven by generative modeling (Shen et al., 2018; Kim et al., 2021; Ren et al., 2021; Jeong et al., 2021; Wang et al.,

---
[*]Equal contribution  [1]X-LANCE Lab, MoE Key Lab of Artificial Intelligence, Jiangsu Key Lab of Language Computing, Shanghai Jiao Tong University [2]Nankai University [3]Microsoft Corporation [4]Shanghai Innovation Institute. Correspondence to: Xie Chen <chenxie95@sjtu.edu.cn>.

*Proceedings of the 43rd International Conference on Machine Learning*, Seoul, South Korea. PMLR 306, 2026. Copyright 2026 by the author(s).

2023), large-scale speech corpora (Kang et al., 2024; He et al., 2024), and increased computational resources. Modern TTS systems (Chen et al., 2024; Ju et al., 2024) can now generate high-fidelity, natural, and expressive speech, enabling broad applications in accessibility, content creation, and human-computer interaction. At the same time, this capability leap represents a double-edged sword, introducing a growing range of ethical and societal concerns. High-fidelity voice cloning lowers the barrier to telecom fraud and disinformation through audio deepfakes. Synthetic speech further threatens biometric authentication, as it can spoof commercial automatic speaker verification (ASV) systems (Wang et al., 2024b). More broadly, synthesizing a person's voice without authorization raises concerns around consent, privacy, ownership, and digital identity (Sharma et al., 2026). In addition, biased training data and narrow evaluation protocols can reinforce societal inequities, leading to uneven quality across demographic groups (Pinhanez et al., 2024) and representational harms such as demeaning portrayals (Michel et al., 2025).

Current TTS evaluation has not kept pace with the expanding complexity and societal reach of TTS technology. Existing practices still center on technical performance in terms of naturalness, intelligibility, speaker similarity, and efficiency, revealing a critical imbalance between technological advancement and evaluation practice. We therefore argue that TTS evaluation must move beyond technical performance to encompass ethical and societal considerations. To this end, we put forward the concept of *Responsible Evaluation* for TTS, structured into three progressive levels that call for a comprehensive rethinking of how evaluation should evolve amid rapid technological progress. **We argue that Responsible Evaluation is essential and urgent for the next phase of TTS development.**

- *Level One: Fidelity and Accuracy.* Evaluation metrics should faithfully reflect a model's true capabilities and limitations.
- *Level Two: Comparability, Standardization, and Transferability.* Evaluation practices should follow scientific rigor to enable meaningful cross-system comparisons.
- *Level Three: Governance, Fairness, and Security.* Evaluation should incorporate ethical and societal implications, aligning TTS development with the public interest and broader principles of responsible AI.

**Contributions** Our contributions to the discourse on TTS evaluation are threefold: (1) *A comprehensive and critical diagnosis of current TTS evaluation practices.* We systematically dissect standard evaluation methodologies across the entire TTS pipeline, covering data, training, inference, and evaluation, and reveal shortcomings in fidelity, transparency, reproducibility, comparability, standardization, transferability, governance, fairness, and security, which collectively hinder genuine progress in TTS technology. (2) *Introduction and elaboration of the concept of Responsible Evaluation.* We propose a three-level concept of Responsible Evaluation that extends beyond the prevailing focus on technical performance to address structural deficiencies in current TTS evaluation and align with broader responsible AI principles. (3) *Actionable recommendations for inspiring future work on responsible evaluation for TTS.* We articulate concrete calls to action for each level of Responsible Evaluation: (i) advancing more robust, discriminative, and comprehensive objective and subjective scoring methodologies; (ii) establishing standardized benchmarks, transparent reporting, and transferable evaluation metrics; and (iii) requiring data provenance disclosure, developing representation-aware benchmarks and protocols, and extending standardized evaluation practices to traceability.

## 2. Background: The Co-evolution of TTS Technologies and Evaluation Methods

Over the past two decades, speech synthesis has undergone a remarkable transformation (Tan et al., 2021a; Xie et al., 2025), evolving from manually crafted statistical models to end-to-end deep learning systems, and more recently to approaches based on diffusion models and large language models (LLMs). Throughout this evolution, subjective evaluation has remained the foundation of TTS assessment. As new capabilities have emerged, such as zero-shot speaker adaptation and fine-grained prosody control, objective metrics have become increasingly important, providing faster, reproducible assessments of specific aspects of synthesis quality and effectively complementing traditional subjective evaluations. As shown in Figure 1, we examine three main phases in the development of TTS technology: the statistical parametric synthesis era, the end-to-end deep learning era, and the era of diffusion models and foundation models. We analyze how evaluation methodologies have evolved alongside advances in model architectures and capabilities.

### 2.1. Statistical Parametric Synthesis Era (2000s)

Building on early rule-driven approaches (Allen et al., 1987; Hallahan, 1995), as well as unit selection concatenative synthesis methods (Moulines & Charpentier, 1990; Hunt & Black, 1996), the early 2000s saw the emergence of Statistical Parametric Speech Synthesis (SPSS) (Yoshimura et al., 1999; Tokuda et al., 2000). These systems model acoustic characteristics of speech such as spectral features, fundamental frequency ($F_0$), and duration using context-dependent HMM (Zen et al., 2009), and later DNN (Zen et al., 2013) and RNN (Zen & Sak, 2015). The generated acoustic parameters are then passed to signal-processing-based vocoders (Kawahara et al., 2001; Morise et al., 2016) that reconstruct the speech waveform. SPSS provides a compact and flexible framework that allows precise control over prosodic elements, including pitch and timing (Zen et al., 2009). This has led to its use in low-resource scenarios and multilingual applications (Zen et al., 2012). However, a major drawback of SPSS is its tendency to produce over-smoothed outputs (Toda et al., 2007), resulting in synthetic speech that sounds dull and lacks natural expressiveness.

In parallel, the evaluation of TTS systems began with modest, informal approaches and has since evolved into standardized, multi-dimensional methodologies. Early research (Tokuda et al., 2000; Yoshimura et al., 1999) primarily relied on visual inspection of spectrograms and pitch contours, alongside informal listening tests, to assess synthesis quality. Subsequently, objective metrics such as mel-cepstral distortion (MCD), $F_0$ root mean square error (RMSE), and voiced/unvoiced classification error were widely adopted to quantitatively evaluate acoustic modeling performance (Toda et al., 2007). Meanwhile, subjective evaluation methods also evolved. Informal listening was gradually replaced by structured AB preference tests (Black & Tokuda, 2005), enabling statistical comparisons between systems based on listener choices. Later, Mean Opinion Score (MOS) evaluations became the standard for capturing absolute judgments of naturalness on a defined scale (King, 2014). These methods are increasingly conducted via crowd-sourcing platforms such as Amazon Mechanical Turk, which allow for large-scale and diverse listener participation (Ribeiro et al., 2011).

### 2.2. Deep Learning End-to-End Era (2016-2021)

Speech synthesis technology entered a transformative era with the rise of fully neural, end-to-end architectures that significantly enhance speech naturalness and simplify the synthesis process. WaveNet (Van Den Oord et al., 2016) generates high-quality raw audio by learning the long-range patterns in sound. Building on this, Tacotron (Wang et al., 2017; Shen et al., 2018) uses attention-based sequence-to-sequence networks to turn text into mel-spectrograms, which are then transformed into waveforms by a neural vocoder. These models eliminate the need for hand-crafted linguistic features and complex alignment procedures, producing speech with more natural prosody and near-human quality. The introduction of Transformer-based models marks a further breakthrough (Li et al., 2019; Ren et al., 2019; 2021). In parallel, diverse generative modeling ap-

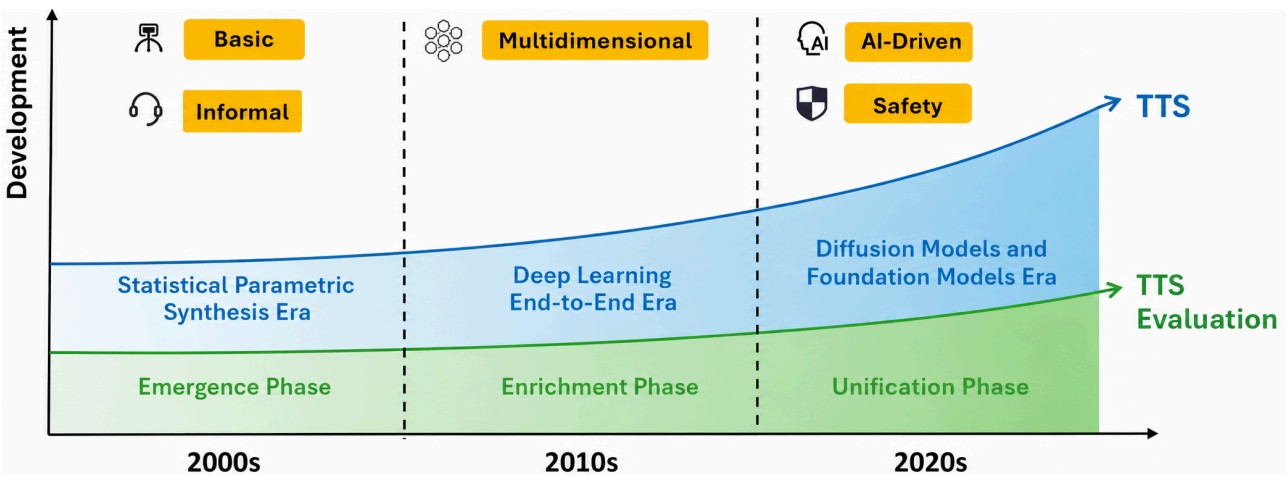

*Figure 1.* Co-evolution of TTS technology and TTS evaluation across three phases.

proaches emerged, including variational (Kim et al., 2021), adversarial (Binkowski et al., 2020), and flow-based models (Miao et al., 2020; Kim et al., 2020), culminating in models like VITS (Kim et al., 2021) that unify acoustic modeling and waveform generation within a single probabilistic framework. These innovations reflect a broader trend toward integrated, data-driven TTS systems capable of capturing the variability and richness of natural speech across diverse speakers, styles, and contexts.

Meanwhile, TTS evaluation practice has gradually evolved to become more comprehensive, centered on subjective assessment, especially MOS, and increasingly supported by diverse objective metrics. MOS became the primary method for evaluating naturalness (Arık et al., 2017; Gibiansky et al., 2017). Comparison MOS (CMOS) (Li et al., 2019; Kim et al., 2020) and Similarity MOS (SMOS) (Chen et al., 2021b) have become common protocols for relative naturalness and speaker similarity, respectively. Objective evaluation gained traction through metrics, including Word Error Rate (WER) for measuring robustness (Ren et al., 2019). As non-autoregressive (NAR) systems emerged, such as FastSpeech (Ren et al., 2019) and Glow-TTS (Kim et al., 2020), inference latency and model efficiency became standard evaluation criteria. Adaptation efficiency (Chen et al., 2021b) also became essential. Then, the evaluation of controllability and diversity entered an exploratory stage. While initial efforts on explicit prosodic modeling are quantified by pitch and energy errors (Ren et al., 2021), systematic evaluation metrics for these aspects remain limited.

## 2.3. Diffusion Models and Foundation Models Era (2022-Present)

The landscape of TTS has been fundamentally transformed by the emergence of generative models (Ramesh et al., 2021; Rombach et al., 2022; Borsos et al., 2023) and LLMs (Ope-

nAI, 2024). TTS systems that leverage powerful sequence modeling have achieved unprecedented generalization, naturalness, and flexibility. Foundation models such as VALL-E (Wang et al., 2023) and its subsequent extensions (Chen et al., 2024; Han et al., 2024; Du et al., 2025a; Yang et al., 2025b; Sun et al., 2025; Yang et al., 2024; Meng et al., 2025; Wang et al., 2025b) redefine TTS as a conditional sequence modeling task over speech tokens, enabling zero-shot capabilities such as voice cloning and style transfer. In parallel, probabilistic generative methods, particularly diffusion models (Shen et al., 2024; Ju et al., 2024) and flow-matching models (Mehta et al., 2024), have advanced the field. Breakthroughs like E2 TTS (Eskimez et al., 2024) and F5-TTS (Chen et al., 2025) demonstrate that these flow-based architectures can achieve high-fidelity, NAR synthesis with simplified alignment. Hybrid approaches such as FELLE (Wang et al., 2025a) and DiTAR (Jia et al., 2025) integrate flow matching into autoregressive (AR) frameworks in token-wise and block-wise manners, respectively, to balance long-range dependency modeling and high-fidelity speech generation. Recently, LLM-based TTS systems have emerged as the dominant paradigm. Models such as CosyVoice 2/3 (Du et al., 2024b; 2025b), Qwen3-TTS (Hu et al., 2026), and OmniVoice (Zhu et al., 2026) are initialized from LLMs (Qwen, 2025a;b) and further extended to support speech generation, leveraging the rich semantic understanding and instruction-following capabilities inherited from text modality to improve synthesis quality and enable style control within a unified foundation model framework. Together, these developments mark a shift toward unified, scalable, and general-purpose TTS systems.

The evaluation of modern TTS systems has increasingly adopted a dual-track framework that combines subjective and objective measures (Wang et al., 2023; Du et al., 2024a; Anastassiou et al., 2024). CMOS and SMOS are now widely

used to assess perceived naturalness and speaker similarity, forming the core of human evaluation protocols. On the objective side, metrics such as WER, speaker embedding similarity (SIM), and model-based predictions of speech quality have become standard practice. Many recent approaches rely on pretrained automatic speech recognition (ASR) models (Radford et al., 2023; Gulati et al., 2020), ASV models (Desplanques et al., 2020), and perceptual quality prediction models (Reddy et al., 2021; Baba et al., 2024) to provide consistent and scalable assessments. This shift reflects a broader evolution toward neural models as evaluators, culminating in the recent adoption of the LLM-as-a-Judge paradigm (Wang et al., 2025e;d; 2026a; Zhang et al., 2025b), which goes beyond scalar scores to deliver interpretable reasoning and fine-grained quality assessments.

However, established evaluation practices remain largely centered on technical quality, leaving ethical and societal implications underexamined. Current protocols rarely require disclosure of training data provenance, licensing conditions, or speaker consent, despite the biometric and personally identifiable nature of voice data. They also risk obscuring fairness concerns: aggregate WER, SIM, and MOS scores can mask degraded quality for underrepresented speech communities, while ASR- and ASV-based metrics inherit disparities from the pretrained models on which they rely (Koenecke et al., 2020; Hutiri & Ding, 2022) and misinterpret minority speech varieties as generation errors. Security risks such as voice impersonation are prominent (Shoaib et al., 2023), yet post-generation traceability is rarely incorporated into standard evaluation practice. Collectively, these gaps motivate the need to extend TTS evaluation beyond technical performance and integrate assessment of governance, fairness, and security into standard evaluation protocols.

## 3. Level One: Ensuring Fidelity and Accuracy in TTS Evaluation

The first level of Responsible Evaluation argues for the necessity of evaluation metrics that faithfully reflect both the perceptual quality of synthesized speech and underlying system performance. When evaluation methodologies are flawed or unreliable, higher-level claims regarding comparability, standardization, or ethical considerations become unfounded. Modern TTS evaluations (Tan et al., 2021b) primarily consider dimensions including naturalness, intelligibility, robustness, speaker similarity, prosody, and system efficiency. These aspects are assessed through a combination of subjective and objective metrics. However, limitations persist in both the effectiveness of these metrics and the comprehensiveness of the evaluation dimensions covered. On the one hand, commonly used metrics sometimes fail to reflect the true capabilities of models, where objective metrics often struggle to align with human percep-

tual judgments (Tee et al., 2026; Yang et al., 2026b), while subjective metrics suffer from methodological inconsistencies (Chiang et al., 2023). On the other hand, the scope of evaluation dimensions remains incomplete (Manku et al., 2025), particularly for complex, real-world scenarios. We elaborate on these issues in the following subsections.

### 3.1. Challenges with Objective Metrics

Objective metrics are valued for scalability and reproducibility, but they face two fundamental limitations. First, the relationship between metric scores and human perception is nonlinear and even non-monotonic, such that improvements in metric values do not necessarily translate into proportional gains in perceived quality. This discrepancy is often attributed to a mismatch between model training objectives and practical evaluation protocols (Wang et al., 2026b). Second, metrics derived from neural models inevitably embody their internal biases and uncertainty (Wang et al., 2024a), rendering evaluation outcomes dependent not only on the input data but also on the metric model itself.

**WER**   To evaluate intelligibility and robustness, WER is computed by comparing ASR transcriptions of synthetic speech with reference texts. While effective at identifying severe intelligibility failures, its reliability is limited in three respects. First, inherent errors in ASR systems (Hsu et al., 2021; Radford et al., 2023) lead to a mismatch between metric scores and actual perceptual quality, even when the synthesized speech is perceptually adequate to humans. Second, WER is not linearly correlated with perceived intelligibility: it focuses on word-by-word accuracy while overlooking whether key information is accurately conveyed (Tee et al., 2026). Third, directly optimizing WER as a reinforcement learning reward signal can be counterproductive. Models optimized for transcription accuracy tend to collapse prosodic variance into monotone output, thereby sacrificing naturalness at the expense of lexical precision (Shin et al., 2026).

**SIM**   To assess speaker similarity, the SIM score is computed by the cosine similarity between speaker embeddings extracted from reference and synthesized speech. These embeddings, derived from speaker verification models like ECAPA-TDNN (Desplanques et al., 2020), can be sensitive to channel variations, background noise, and even phonetic content, leading to unstable scores. More fundamentally, these models are trained with discriminative objectives for speaker identity classification, which are misaligned to quantify continuous perceptual similarity. In practice, once the SIM score exceeds a certain threshold, further improvements offer limited perceptual gains (Wester et al., 2016).

**Predicted MOS**   Predicted MOS scores are generated by models trained on human ratings (Cooper & Yamag-

ishi, 2021; Liu et al., 2025) collected following ITU-T P.808 (Naderi & Cutler, 2020). While these models offer a scalable alternative to human evaluation, they struggle with generalization and uncertainty estimation, primarily due to limitations in the diversity of training data and model representational power. Prior works (Wang et al., 2025c; Cooper et al., 2022) have shown that existing MOS prediction models often produce inconsistent results even on in-domain data, and their performance degrades significantly when applied to out-of-domain data. A typical example of domain mismatch is the widespread use of DNSMOS (Reddy et al., 2021; Cumlin et al., 2024; Reddy et al., 2022), which is trained on speech enhancement data yet commonly employed to evaluate synthesized speech. Moreover, MOS prediction models generally lack uncertainty estimation (Wang et al., 2024a), as they typically provide only point estimates without associated confidence intervals, making it difficult to assess the reliability of the predicted quality scores. This remains rarely examined in current research.

**F0** To assess speech prosody, current evaluation practices commonly employ log $F_0$ RMSE aligned via dynamic time warping (DTW) (Galdino et al., 2025). However, this approach is fundamentally limited in its ability to capture the multidimensional nature of prosody as it only captures pitch while ignoring other essential constituents, including rhythm, stress, and intensity (Arvaniti, 2020). Furthermore, this metric has been shown to correlate weakly with human perceptual judgments (Yang et al., 2026b).

### 3.2. Challenges with Subjective Metrics

Subjective evaluation remains the primary choice for assessing perceptual quality in TTS, with MOS serving as the dominant protocol. MOS employs a five-point absolute category rating scale to rate individual utterances. Alternative protocols such as CMOS and MUSHRA are used for pairwise or comparative assessments. Although broadly regarded as the gold standard, these methods fall short in terms of sensitivity, consistency, and practical feasibility. One major drawback of MOS stems from its limited resolution. As the quality of synthetic speech continues to improve, MOS scores tend to saturate (Wang et al., 2025f). This ceiling effect obscures perceptual differences across high-performing systems, making it increasingly difficult to distinguish among them and judgments sensitive to listener bias and preference. Another issue arises from the inherent variability in subjective ratings. Factors such as listener bias, contextual framing, playback conditions, and even day-to-day mood can introduce substantial noise. Without rigorous rater calibration and experimental controls, evaluations become unreliable. Moreover, the high cost associated with subjective evaluations presents a practical barrier. The process of recruiting a large and diverse pool of listeners, along

with the need to ensure controlled testing conditions, demands considerable time and resources. These requirements often limit the feasibility and scale of such evaluations.

### 3.3. Underexplored Dimensions in TTS Evaluation

Existing evaluation dimensions in TTS fail to keep pace with the growing complexity of real-world applications. Widely used metrics capture only a narrow portion of what matters in practical synthesis scenarios (Manku et al., 2025). We therefore identify the following key evaluation dimensions that are essential for forward-looking assessment.

**Mathematical Symbols and Formulas** Modern TTS systems like Qwen3-TTS (Hu et al., 2026) are deployed in educational, scientific, and accessibility-oriented scenarios, where accurate verbalization of mathematical symbols, formulas, and structured notations is critical. Mathematical expressions often exhibit non-linear and deeply nested structures, implicit grouping, and context-dependent reading conventions that are poorly handled by current text normalization pipelines. Errors in symbol pronunciation, operator scope, or structural cues can severely impair comprehension, yet often remain invisible to ASR-based evaluations. Beyond mathematics, real-world scenarios often interleave formulas with diverse content, further complicating evaluation. While recent efforts such as EmergentTTS-Eval (Manku et al., 2025) begin to cover emails, phone numbers, URLs, addresses, STEM equations, units, and notations, the community still lacks multi-domain benchmarks and evaluation protocols that systematically assess symbolic and structured speech synthesis, leaving a gap between real-world requirements and current evaluation practices.

**Long-form Synthesis** In real-world applications such as audiobooks and podcasts, coherence across sentences and stability in prosody and speaker identity are essential. However, most existing evaluations center on short utterances such as LibriTTS (Zen et al., 2019) and Seed-TTS-eval (Anastassiou et al., 2024). There is a lack of representative test sets and metrics specifically designed to assess long-form fluency, consistency, and discourse-level control.

**Emotional Expressiveness** Recent TTS models have demonstrated increasing capability in synthesizing expressive speech (Du et al., 2024b; Hu et al., 2026), yet evaluation methodologies remain underdeveloped. In particular, there is no consensus on emotion taxonomies or scales for emotion intensity, and subjective metrics like emotion MOS often lack sensitivity to subtle distinctions (Yang et al., 2025a). Moreover, widely used emotional speech datasets (Busso et al., 2008; Livingstone & Russo, 2018; Cao et al., 2014) primarily rely on discrete labels and provide limited coverage of expressive diversity.

**Punctuation Sensitivity** Punctuation plays a vital role in shaping prosody by guiding pauses, emphasis, and intonation contours. However, current evaluation practices often overlook whether synthesized speech appropriately reflects punctuation cues in the input text. There is a lack of established metrics to quantify punctuation sensitivity or its impact on perceived fluency and naturalness.

### 3.4. Recommendations

To promote fidelity and accuracy in TTS evaluation, we propose the following actionable recommendations, grounded in a reevaluation of current evaluation practices:

- **Interpreting objective metrics reliably.** Objective score differences should be interpreted with caution, given non-linear scaling, diminishing returns, domain-specific biases, and prediction uncertainty. We advocate reporting uncertainty estimates for model-predicted MOS, especially under out-of-distribution conditions. Without uncertainty estimates, minor differences in predicted MOS should not be interpreted as genuine performance gains.

- **Developing discriminative evaluation protocols.** We encourage the development of evaluation protocols that remain sensitive even when modern TTS systems approach human-level naturalness. Subjective methods such as the audio Turing test (Wang et al., 2025f) can mitigate score saturation and improve interpretability, while objective metrics should move beyond word-level correctness to assess key information preservation (Tee et al., 2026).

- **Expanding evaluation to real-world capabilities.** We advocate for a broader evaluation scope that reflects real-world TTS use cases, including long-form coherence, emotional expressiveness, and faithful rendering of complex content such as mathematical expressions.

## 4. Level Two: Ensuring Comparability, Standardization, and Transferability in TTS Evaluation

The second level of Responsible Evaluation builds upon the foundation of fidelity and accuracy established in the first level, arguing the importance of scientific rigor to enable meaningful cross-system comparisons. Without standardized practices, even technically valid assessments fail to support meaningful comparison or generalizable conclusions. Current evaluation practices in TTS research remain fragmented, characterized by inconsistent methodologies, limited transparency, and poor transferability in metrics.

### 4.1. Challenges with Inconsistent Evaluation Practices

**Evaluation Datasets** A primary challenge to comparability stems from the inconsistent usage of evaluation datasets.

The most commonly used test set, LibriSpeech (Panayotov et al., 2015) test-clean, is employed in divergent ways across various TTS studies. For example, VALL-E (Wang et al., 2023) utilizes 1234 utterances for zero-shot evaluation, while NaturalSpeech 3 (Ju et al., 2024) and MaskGCT (Wang et al., 2024c) employ only 40 utterance subsets, and F5-TTS (Chen et al., 2025) uses 1127 utterances with punctuation and capitalization. Such disparities in test set size significantly influence evaluation metrics like WER, as detailed in Appendix A, making cross-study comparisons unreliable. Moreover, most TTS studies do not release prompt speech lists. However, the sequence of prompt speech can impact performance, making results difficult to reproduce.

**Inference Tasks** Inference tasks to evaluate zero-shot TTS are also fragmented. VALL-E (Wang et al., 2023) introduced two tasks, *Continuation*, which uses the first three seconds of an utterance as a prompt and continues the speech, and *Cross-Sentence*, which prompts with a full utterance from the same speaker. However, later work such as E2 TTS (Eskimez et al., 2024) redefines the *Continuation* task by using the last three seconds of a truncated segment as the prompt. These inconsistencies in task definition lead to incomparable evaluation results across different works.

**SIM** The computation of SIM scores varies across studies. SIM-o measures the similarity between the synthesized speech and the original prompt, while SIM-r measures the similarity between the synthesized speech and the reconstructed prompt. SIM-r is not comparable across systems using different reconstruction methods. Evaluation practices for SIM-o also differ: VALL-E (Wang et al., 2023) excludes the prompt segment when computing similarity, while VALL-E 2 (Chen et al., 2024) includes the prompt. As detailed in Appendix B, these differences lead to incomparability across works.

**MOS** Widely adopted MOS evaluations frequently depart from recommended standards. While ITU-T P.808 (Naderi & Cutler, 2020) provides detailed protocols for conducting listening tests, many studies refer to MOS without reporting essential details, including rating scale definitions, rater calibration, playback conditions, and whether listeners rated naturalness or overall quality. Such inconsistencies reduce the reliability and comparability of MOS scores.

**Text Preprocessing** Text preprocessing introduces another variation. Differences in text normalization, phonemization, and treatment of polyphonic words can affect synthesis quality, thus undermining the strict comparability of reported results across different studies.

## 4.2. Challenges with Transparency in Evaluation Reporting

**RTF**   While Real-Time Factor (RTF) is the standard efficiency metric in TTS, its reporting frequently lacks critical details such as hardware configuration, batch size, length of prompt speech, and whether inference is performed in streaming mode. These omissions hinder reproducibility and cross-system comparability. This ambiguity is particularly problematic for NAR systems, where the length of prompt speech significantly affects RTF yet is rarely reported. Additionally, some studies exclude components such as vocoders or speech detokenizers when computing RTF, thereby misrepresenting the actual end-to-end latency of the synthesis pipeline.

**MOS**   The reporting of MOS in TTS research often lacks transparency (Wang et al., 2025f). Despite the importance of standardized reporting in human evaluations, many TTS studies underreport details of testing methodologies. Information regarding listener recruitment, screening procedures, compensation, and the evaluation interface is often omitted, which complicates the assessment of result replicability.

## 4.3. Challenges with Metric Transferability

**SIM**   The computation of SIM requires access to reference speech, which limits its applicability in horizontal comparisons across different TTS research. External evaluators often lack access to the original reference speech, hence are unable to directly compare the newly generated speech to previous ones, further hindering the transferability of this metric across studies.

**MOS**   MOS evaluations are not transferable across studies. Direct comparisons of MOS scores across studies are meaningless due to the subjective nature of MOS (Kirkland et al., 2023). Instead, any new comparison requires both new and previously generated speech to be jointly re-evaluated within the same subjective listening test.

## 4.4. Recommendations

To advance comparability, standardization, and transferability in TTS evaluation, we propose the following actionable recommendations:

- ***Distinguishing comparable from incomparable results.*** Scores derived from different datasets, tasks, or configurations should not be treated as interchangeable. Any protocol deviations should be reported explicitly to avoid misleading comparisons.
- ***Adhering to standardized evaluation protocols.*** When formal standards such as ITU-T P.808 for MOS are available, researchers should adhere to them consistently. In the absence of formal standards, alignment with widely adopted practices is encouraged to promote practical convergence across studies.
- ***Reporting evaluation details transparently.*** Evaluation reports should disclose details, including but not limited to dataset splits, prompt lists, inference task definitions, metric configurations, human listening test procedures for MOS, and measurement setups for RTF.
- ***Developing transferable metrics.*** Model-based evaluation, including recent LLM-as-a-Judge approaches, offers a scalable alternative to human evaluation. We encourage the development of human-aligned automatic metrics (Yang et al., 2026b;a) that produce transferable scores under shared evaluation conditions, enabling cross-system comparison without repeated human re-evaluation.

# 5. Level Three: Ensuring Governance, Fairness, and Security in TTS Evaluation

The third and also the broadest level of Responsible Evaluation centers on the ethical and societal implications of TTS technology. While technical fidelity and scientific comparability form the foundation of sound evaluation, they are insufficient for assessing TTS technology as a sociotechnical system (Selbst et al., 2019) whose data, models, and outputs can affect identity, consent, representation, and security in real-world human–computer interaction. As TTS systems grow more realistic and accessible, concerns related to governance (Sharma et al., 2026), fairness (Michel et al., 2025), and security (Wang et al., 2024b) have drawn growing attention. These concerns underscore the urgent need to move beyond purely technical performance indicators and explicitly incorporate ethical and societal implications into TTS evaluation. Current evaluation practices often overlook these dimensions, leaving the broader consequences of TTS development and deployment insufficiently scrutinized.

## 5.1. Challenges with Governance: Data Legitimacy, Consent, and Accountability

Governance concerns arise from the fact that voice is not merely acoustic data, but a personally identifiable and biometric signal. Many modern TTS systems are trained on large-scale speech datasets (Ma et al., 2024; He et al., 2024; Chen et al., 2021a; Yang et al., 2025c) comprising vast amounts of voices collected from the internet, where speaker consent, licensing terms, and data provenance are often unclear or difficult to verify at scale.

Current TTS evaluations rarely treat data governance as part of the evaluation protocol. Some studies (Zhang et al., 2025a), especially technical reports, describe training data using vague terms such as "in-house data" without disclosing the sources, licenses, or collection procedures of the

voices. Together, these practices create a data legitimacy issue: even when a model produces high-quality synthetic speech, its development may rely on voice data whose authorization remains unclear, exposing developers to potential legal risks. Responsible Evaluation should therefore assess not only synthetic speech quality, but also whether the training data are transparently documented, properly licensed, and authorized for the intended TTS use case.

### 5.2. Challenges with Fairness: Disparities and Representational Harms

Fairness in TTS evaluation concerns not only average perceptual quality, but also whether synthetic voices equitably represent and preserve diverse speech communities. Aggregate evaluation scores can mask degraded synthesis quality, reduced intelligibility, or weakened identity preservation for underrepresented linguistic and demographic groups. Such disparities further lead to representational harms, where underrepresented voices are stereotyped (Ovacık, 2025; Puhach et al., 2025), demeaned (Michel et al., 2025), or homogenized (Prinos et al., 2024).

A key challenge is that current evaluation protocols often treat naturalness, speaker similarity, and listener preference as socially neutral criteria, even though these judgments can be shaped by the backgrounds of raters. Automatic evaluators also introduce bias: ASR-based intelligibility metrics inherit recognition disparities across speech varieties (Koenecke et al., 2020), while ASV-based speaker similarity metrics inherit biases from ASV systems (Hutiri & Ding, 2022). Responsible Evaluation should therefore include group-disaggregated reporting, rater-background documentation where applicable, and audits of human and automatic evaluators for group-specific bias.

### 5.3. Challenges with Security: Misuse, Spoofing, and Traceability

Voice impersonation is among the most immediate societal risks raised by modern TTS, as high-fidelity synthesis can enable realistic imitation of a person's voice. The growing availability of open-source models and API-based services further lowers the barrier to misuse, including telecom fraud (Zhang et al., 2025c), misinformation and disinformation via deceptive media (Shoaib et al., 2023), and spoofing biometric authentication systems (Wang et al., 2024b). However, such malicious-use scenarios remain largely outside standard TTS evaluation practices, revealing a gap between technical evaluation and security-aware evaluation.

Traceability is a key requirement for secure TTS deployment. When synthetic speech cannot be reliably detected or traced, high-fidelity TTS weakens trust in voice-mediated communication and complicates accountability after deployment. Responsible Evaluation should therefore examine whether TTS systems support traceability mechanisms (Wen et al., 2025; Zhou et al., 2024) that enable post-generation detection, attribution, and accountability.

### 5.4. Recommendations

To promote governance, fairness, and security in TTS evaluation, we propose actionable recommendations as follows:

- *Mandating disclosure of training data provenance.* Evaluation reports should move beyond vague terms such as "in-house data" by specifying data sources, licenses, consent conditions, and collection procedures to ensure verifiable transparency and accountability.

- *Constructing representation-aware benchmarks and protocols.* We encourage the development of evaluation benchmarks that cover diverse speech communities, with group-disaggregated reporting across key metrics. We also encourage the development of representation-aware automatic evaluators, such as multilingual ASV models for speaker-similarity assessment, instead of assuming English-centric models generalize across languages, accents, and speaker groups (Chen et al., 2022).

- *Extending standardized evaluation to traceability.* TTS systems are encouraged to adopt traceability mechanisms such as imperceptible watermarking (Zhao et al., 2025). Standardized TTS evaluation practices should therefore be extended to assess whether synthetic speech can be reliably detected and traced after generation.

## 6. Alternative Views

Our perspectives are intended to stimulate further discussion. While we acknowledge diverse viewpoints, we discuss several alternative views to our position below:

**Alternative View 1: Concerns about Increased Evaluation Complexity.** Some practitioners caution that introducing additional evaluation metrics could complicate the evaluation process, particularly in industrial contexts where scalability and efficiency are critical. They also note that an overabundance of criteria might risk fragmenting TTS evaluation practices, thereby reducing comparability and standardization. We believe that while expanding evaluation dimensions and introducing new metrics may pose short-term challenges, such efforts are essential to ensure that TTS evaluation evolves in step with technological advances and real-world requirements. As in many areas of technology, development often shifts from diversification to convergence, ultimately leading to unified, stable practices.

**Alternative View 2: Balancing Rapid Progress with Legal and Ethical Considerations.** Some practitioners caution that excessive emphasis on legal and ethical aspects

could inadvertently slow technological innovation. In particular, overly restrictive interpretations of data copyright may constrain progress in low-resource languages and domains where available data are scarce. We acknowledge that, for low-resource speech technologies, uneven copyright awareness and the scarcity of high-quality data present genuine challenges to TTS development. However, these challenges are not insurmountable. Doctrines such as Fair Use can provide limited flexibility in some jurisdictions for responsible data use. Nevertheless, this remains a limitation of strict governance-oriented evaluation, suggesting the need for practical transparency and accountability mechanisms.

## 7. Conclusion

As TTS technology continues to advance, current evaluation practices have become increasingly inadequate for capturing the full range of capabilities, limitations, and societal impacts of modern TTS systems. In response to this urgent need, we introduce the concept of Responsible Evaluation, structured around three progressive levels. At the first level, we advocate the reevaluation of current evaluation practices to faithfully and accurately reflect a model's true capabilities and limitations through more robust, discriminative, and comprehensive objective and subjective scoring methodologies. At the second level, we call for the adoption of standardized benchmarks and protocols that support meaningful comparisons and ensure reproducibility across models and studies. At the third level, we emphasize the importance of integrating governance, fairness, and security considerations throughout the evaluation pipeline. We believe that embracing Responsible Evaluation is not only essential for advancing scientific progress in TTS but also critical for guiding TTS development in alignment with broader societal interests and responsible AI principles.

## Acknowledgements

This work was supported by the National Natural Science Foundation of China (No. U23B2018), Shanghai Municipal Science and Technology Major Project under Grant 2021SHZDZX0102, and Yangtze River Delta Science and Technology Innovation Community Joint Research Project (2024CSJGG1100).

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

## A. Case Study on Variants of LibriSpeech *test-clean* Subsets

Multiple versions of the LibriSpeech *test-clean* subset are used across recent TTS works, which leads to inconsistencies in reported results. One version contains 1234 utterances and is used in systems such as VALL-E (Wang et al., 2023), VALL-E 2 (Chen et al., 2024), MELLE (Meng et al., 2025), and PALLE (Yang et al., 2025b). Another version contains 40 utterances and is used in works including NatureSpeech 3 (Ju et al., 2024) and MaskGCT (Wang et al., 2024c). Other subsets, such as the one used in F5-TTS (Chen et al., 2025), also exist. These differences cause substantial variation in WER evaluations even for the same model.

To demonstrate this issue, we evaluate the open-sourced MaskGCT[1] on two commonly used variants of the *test-clean* subset. WER is computed between ASR transcription of synthesized audio and the ground-truth text, using the HuBERT-Large ASR model[2] (Hsu et al., 2021). The WER differs significantly across the two versions, ranging from 2.63 to 4.22, as shown in Table 1. This observation argues the importance of clearly reporting dataset versions and evaluation protocols to ensure fair and reproducible comparisons.

*Table 1.* WER of MaskGCT for the cross-sentence task on different variants of the LibriSpeech *test-clean*.

| Subset Variant | WER (%) |
|---|---|
| 40 utterances (Wang et al., 2024c) | 2.63 |
| 1234 utterances (Yang et al., 2025b) | 4.22 |

## B. Case Study on Inconsistencies in SIM-o Evaluation Protocols

SIM-o is defined as the cosine similarity between speaker embeddings extracted from original speech and synthesized speech. Commonly, SIM-o is computed using WavLM-TDNN[3] (Chen et al., 2022), where the score ranges within $[-1, 1]$, with higher values indicating greater speaker similarity.

However, there are two practices for computing SIM-o for the continuation task. One approach, adopted by VALL-E (Wang et al., 2023), computes speaker similarity between the first 3-second ground-truth speech prompt and the remaining synthesized speech, excluding the prompt. Alternatively, another approach, as used in VALL-E 2 (Chen et al., 2024), computes the similarity between the full synthesized speech, including the prompt and the entire ground-truth speech.

Table 2 illustrates this difference using a representative case. These practices result in substantial differences in SIM-o scores, with an absolute value difference of up to 0.151. This case argues the necessity of clearly specifying the SIM-o computation method when reporting speaker similarity results for the continuation task.

*Table 2.* SIM-o scores with or without prompt for the continuation task on the LibriSpeech *test-clean*.

| Protocol | SIM-o |
|---|---|
| Without Prompt (Wang et al., 2023) | 0.754 |
| With Prompt (Chen et al., 2024) | 0.905 |

---

[1] https://huggingface.co/amphion/MaskGCT
[2] https://huggingface.co/facebook/hubert-large-ls960-ft
[3] https://github.com/microsoft/UniSpeech/tree/main/downstreams/speaker_verification#pre-trained-models

