# OpenReview forum: "Position: Towards Responsible Evaluation for Text-to-Speech"
_ICML.cc/2026/Position_Paper_Track — ICML 2026 Position Paper Track regular_

### Official Review · Reviewer_N9Uj · 2026-02-23

**Significance:** 3
**Argument Clarity:** 2
**Rating:** 4
**Confidence:** 4

**Questions:**

Please address my concerns listed above. My final rating may be revised based on the authors' rebuttal and the following discussion period.

**Alternative Views Section:**

Yes

**Compliance With Llm Reviewing Policy A Conservative:**

Affirmed.

**Discussion Potential:**

3

**Final Justification:**

Although other reviewers' reactions are unanimously positive to the paper, I personally don't have strong preferences on whether this paper should be accepted (or not). The Text-to-Speech task is out of my research scope. To respect the authors' effort on the new comparison table, I raise my score to borderline accept.

**Paper Summary:**

This paper proposes the position of responsible evaluation for modern TTS from three levels: (1) accuracy, (2) standardization, and (3) ethics. The evidences are from current evaluation failures (metric misrepresentation, MOS saturation, inconsistent tasks) and misuse risks (e.g., deepfake, biometric spoofing).

**Position:**

Yes

**Position In Title:**

Yes

**Related Work:**

2

**Strengths And Weaknesses:**

**Strength**
1. The paper is well-written with a clear hierarchy and concrete subclaims per category.
2. The research topic of responsible evaluation for high-fidelity generative systems is timely and important.
3. The reasoning from evidence to claims is fair.

**Weakness**
1. The proposed position is too general and not very distinctive. It vaguely argues a responsibility paradigm that existing surveys [1,2,3] have already emphasized more thoroughly, such as the multi-objective alignment (e.g., fairness, privacy, security, distribution shift).

2. The paper lists many metric problems and evaluates inconsistencies as evidences. However, it does not sufficiently state how the proposed position will revise these metrics quantitatively.

[1] Ji et al., AI Alignment: A Comprehensive Survey. https://arxiv.org/abs/2310.19852

[2] Yang et al., Reliable and Responsible Foundation Models. https://openreview.net/forum?id=nLJZh4M6S5

[3] Huang et al., On the Trustworthiness of Generative Foundation Models: Guideline, Assessment, and Perspective. https://arxiv.org/abs/2502.14296

**Support:**

2

---

> ### Author Rebuttal · Authors · 2026-03-29
>
> Thank you for providing your valuable and constructive feedback! Please find below the responses to each comment.
>
> ---
>
> **[W1]**: *"The proposed position is too general and not very distinctive. It vaguely argues a responsibility paradigm that existing surveys [1,2,3] have already emphasized more thoroughly..."*
>
> Thanks for your comment. We agree that responsibility in AI has been extensively discussed in prior surveys [1,2,3]. However, we respectfully argue that **our paper addresses a different and underexplored problem: the reliability and adequacy of evaluation itself**.
>
> **Existing surveys focus on how to build or assess responsible models, implicitly assuming that evaluation itself is trustworthy. Our work challenges this assumption and argues that current evaluation remains insufficient to capture the full capabilities, limitations, and risks of modern TTS systems.**
>
> We clarify the distinction below:
>
> |                 | [1] AI Alignment Survey                                    | [2] Reliable and Responsible FMs Survey                 | [3] Trustworthiness of GenFMs                                | Ours                                                         |
> | --------------- | ---------------------------------------------------------- | ------------------------------------------------------- | ------------------------------------------------------------ | ------------------------------------------------------------ |
> | **Paper Type**  | Survey (descriptive: what have been done)                  | Survey                                                  | Survey + Benchmark                                           | Position Paper (normative: what should be done)              |
> | **Domain**      | General AI                                                 | FMs (LLMs, MLLMs, Image GMs, Video GMs)                 | GenFMs (T2I, LLMs, VLMs)                                     | TTS-specific                                                 |
> | **Focus**       | Build aligned AI systems                            | Build reliable and responsible foundation models | Assess trustworthiness of GenFMs                      |  TTS responsibly evaluation |
> | **Perspective** | System design, training, and assurance                     | Risk mitigation and governance                          | Benchmarking framework                                       | Meta-evaluation (evaluating the evaluation)                  |
> | **Approach**    | Top-down: RICE framework → categorize alignment techniques | Top-down: nine dimensions → review solutions   | Top-down: governance policies→ principles → benchmark | Bottom-up: TTS evaluation practices failures → three-level concept of responsible evaluation → recommendations |
>
> Beyond differences in scope and format, the key novelty of our work lies as follows:
>
> - **Meta-evaluation as the missing link from principles to practice.**
> We identify a critical but overlooked gap between high-level responsibility principles and their practical realization: *the reliability of evaluation itself*. Without scrutinizing the evaluation, claims about responsibility cannot be reliably grounded.
>
> - **Bottom-up diagnosis grounded in TTS practice.**
>   Rather than a top-down survey, we identify concrete shortcomings across TTS practices that are not explicitly examined in existing higher-level surveys.
>
> - **Actionable framework.**
>   As a position paper, we focus on *what should be done* rather than merely surveying *what has been done*. We introduce a three-level concept of Responsible Evaluation, and derive concrete calls to action for each level that directly inspire new metrics, benchmarks, and evaluation protocols.
>
> ---
>
> **[W2]** *"...it does not sufficiently state how the proposed position will revise these metrics quantitatively."*
>
> Thanks for your comment. We would like to clarify that, **as a position paper, our goal is not to propose new quantitative metrics, but to identify systematic limitations in current evaluation practices and to provide a structured concept of Responsible Evaluation that can guide future metric and benchmark design.** Specific quantitative metric revisions are therefore downstream work informed by these principles.
>
> At the same time, we agree that the actionable implications can be made more concrete. **In the revision (with an extra page), we will further elaborate on how our position translates into practical evaluation improvements.** For example, evaluation of predicted MOS should explicitly account for uncertainty. Instead of reporting only point estimates such as `Model A: 4.32 vs. Model B: 4.35`, we advocate uncertainty-aware reporting such as `Model A: 4.32 ± 0.08 vs. Model B: 4.35 ± 0.09`, indicating that the two models are not statistically distinguishable.

---

> > ### Author Rebuttal · Reviewer_N9Uj · 2026-04-06
> >
> > Fairly speaking, the rebuttal has addressed my concerns. I appreciate the comparison table, which is readable and clear. I suggest that authors include it in the final version to better justify the contribution. The text-to-speech task is not fully within my research scope. To respect the authors' valuable effort on the great rebuttal, I raise the rating.

---

### Official Review · Reviewer_ir3g · 2026-03-12

**Significance:** 3
**Argument Clarity:** 4
**Rating:** 5
**Confidence:** 4

**Questions:**

- Are the authors considering this point broadly across TTS, including language agnostic evaluation, etc.?
- To what degree is the target audience the TTS community? I am wonder if the call to action is broader and should bring multi-disciplinary researchers together to solve some of the issues more fundamentally? For example, speech scientists (e.g., with expertise in articulatory phonetics), linguists, psychologists, etc., to bring a independence and balance to the discussion?

**Alternative Views Section:**

Yes

**Compliance With Llm Reviewing Policy A Conservative:**

Affirmed.

**Discussion Potential:**

3

**Paper Summary:**

The authors present the argument that there are broad issues with how text-to-speech systems are evaluated, and that evaluation has not kept pace with the development of the technology. The shortcomings can led to abuse of systems and a mis-trust of reported evaluations. Instead the authors call for Responsible Evaluation that: 1) use metrics that are more faithful to capabilities, 2) standardized protocols for evaluation to ensure fair and reliable reporting, and 3) ethical and risk oversight to consider societal implications of TTS evaluation (e.g., provenance and copyright of data, and so on).

**Position:**

Yes

**Position In Title:**

Yes

**Related Work:**

3

**Strengths And Weaknesses:**

**Strengths**
- The position is laid out clearly throughout the paper.
- There are clear calls to action and recommendations outlined for each section.
- The supporting evidence provided in the appendix nicely supports some of the issues highlighted.

**Weakness**
- The scope is narrow. This is a problem for TTS, but one could argue it is a much broader problem. A more ambitious position paper might have considered this more broadly.
- There is precedent for overcoming many of the issues put forward in this paper. For example working groups like MPEG have overcome exactly many of the issues put forward. Understanding why the TTS community have not adopted such standards is not clear.

I highlight the scope as a weakness, but I concede that it also has allowed the authors to more clearly focus arguments towards specifics of TTS.

**Support:**

4

---

> ### Author Rebuttal · Authors · 2026-03-30
>
> We sincerely thank you for providing your constructive and positive feedback! Please find below the responses to each comment.
>
> ---
>
> **[W1]** *"The scope is narrow. This is a problem for TTS, but one could argue it is a much broader problem. A more ambitious position paper might have considered this more broadly."*
>
> Thanks for your point. We agree that many issues discussed extend beyond TTS to broader omni foundation models and responsible AI evaluation. Our focus on TTS is intentional:
> - TTS is a mature domain now facing concrete and escalating challenges, allowing us to ground the discussion in a setting where gaps have immediate real-world consequences and yield actionable insights.
> - As voice interaction becomes increasingly central to human-computer interaction, leading foundation models (e.g., GPT-4o, Gemini) are integrating speech generation as a core capability, making responsible evaluation of speech systems directly relevant to the broader ML community.
> - Concerns such as audio deepfakes and voice cloning have escalated into mainstream ML challenges, actively discussed at NeurIPS, ICML, and by policymakers worldwide.
>
> We therefore view TTS not as a narrow application domain, but as a representative and tractable entry point for addressing broader evaluation challenges, where sufficient specificity exists to make the position both well-supported and impactful.
>
> ---
>
> **[W2]** *"There is precedent for overcoming many of the issues put forward in this paper. For example, working groups like MPEG have overcome exactly many of the issues put forward. Understanding why the TTS community has not adopted such standards is not clear."*
>
> We appreciate this observation. Indeed, efforts such as MPEG demonstrate that many evaluation and standardization challenges can be addressed through sustained community coordination. However, **unlike audio compression, which measures signal fidelity against an absolute ground-truth reference, TTS encompasses heterogeneous tasks (e.g., zero-shot cloning, instructed, expressiveness, and multilingual).** Combined with the rapid pace of TTS development, this heterogeneity has left evaluation practices highly fragmented. As discussed in Level 2, inconsistencies persist across datasets, tasks, prompts, similarity protocols, and MOS reporting. Critically, awareness of evaluation inconsistencies within the TTS community remains inadequate. Even in very recent top-tier publications (ICML, NeurIPS, ACL), TTS evaluation [1, 2, 3] still exhibits recurring problems:
> - Different SOTA systems claim SOTA on different LibriSpeech subsets.
> - Comparison tables mix results from vastly different evaluation sizes (e.g., 40 vs. 1,234 samples).
> - Baseline results cited in tables are not derived from the same evaluation set as in the original work.
> - Limited reporting of subjective evaluation protocols raises concerns about possible cherry‑picking.
>
> The persistence of these issues is precisely what motivates our position paper.
>
> **References**:
>
> [1] DiTAR: Diffusion Transformer Autoregressive Modeling for Speech Generation. ICML 2025
>
> [2] F5-TTS: A Fairytaler that Fakes Fluent and Faithful Speech with Flow Matching. ACL 2025
>
> [3] MaskGCT: Zero-Shot Text-to-Speech with Masked Generative Codec Transformer. NeurIPS 2025
>
> ---
>
> **[Q1]** *"Are the authors considering this point broadly across TTS, including language agnostic evaluation, etc.?"*
>
> Yes. While the current draft does not explicitly center on multilingual evaluation, the proposed Responsible Evaluation concept is not limited to a single language. In particular, Level 3 emphasizes the importance of fairness-oriented evaluation datasets covering diverse accents, genders, and languages.
>
> We will revise the paper to more explicitly highlight this broader scope.
>
> ---
>
> **[Q2]** *"To what degree is the target audience the TTS community? I am wonder if the call to action is broader and should bring multi-disciplinary researchers together to solve some of the issues more fundamentally? For example, speech scientists (e.g., with expertise in articulatory phonetics), linguists, psychologists, etc., to bring independence and balance to the discussion?"*
>
> Our primary focus is on the TTS community, as many practical recommendations on evaluation protocols, benchmarks, and evaluation practices need to be initiated within it. However, we do not view this effort as confined to TTS; rather, we see it as a natural entry point for broader, cross-disciplinary collaboration. As TTS evaluation increasingly involves aspects such as human perception, prosody, bias, privacy, security, and potential misuse, expertise from multiple fields becomes essential. This includes speech science (e.g., articulatory phonetics), linguistics, psychology, AI safety, and ethics.
>
> We will revise the paper to further emphasize this interdisciplinary perspective.

---

> > ### Author Rebuttal · Reviewer_ir3g · 2026-04-03
> >
> > Thanks for the responses to my questions. I will continue to recommend accepting the paper. None of the questions/concerns I raised were blockers, rather I think they could make the position more impactful.

---

### Official Review · Reviewer_Vzdz · 2026-03-12

**Significance:** 2
**Argument Clarity:** 3
**Rating:** 4
**Confidence:** 3

**Questions:**

NA

**Alternative Views Section:**

Yes

**Compliance With Llm Reviewing Policy A Conservative:**

Affirmed.

**Discussion Potential:**

3

**Final Justification:**

Most of the less important weaknesses have been addresses satisfactorily in the rebuttal. The main weakness in my view still remains: the stated position is overly broad and lacks focus.

**Paper Summary:**

The paper argues that TTS evaluation practices suffer from several drawbacks and limitations. It provides an overview of TTS evaluation approaches and how they relate to evolving technologies and then lists and discusses problems with them. The paper advocates for a number of best practices which would improve on the usual approach to evaluation of TTS systems, and relates these to three levels: (1) fidelity and accuracy, (2) comparability, standardization and transferability and (3) ethical and risk overview. The appendix provides three simple case studies illustrating inconsistencies and lack of validity in TTS evaluation practices.

**Position:**

No

**Position In Title:**

No

**Related Work:**

3

**Strengths And Weaknesses:**

The paper does not really state a concrete, well-defined position. Instead it argues for a number of losely related improvements in evaluation practices of TTS systems. This lack of focus on a single well-delimited issue is the main weakness of this paper.

Another weakness is that most of the problems with the evaluation are relatively obvious, and the main reason they are not being addressed is that addressing them is costly and involves tradeoffs.

The recommendations in 5.5 related to ethical oversight are rather generic and lack more in depth discussion.

Figure 1 seems too handwavy to be useful in a scientific venue. I'm not sure what the introduction of the different phases adds to the discussion.

Beside the above weaknesses, most of the critiques of evaluation practices are reasonably well argued and supported.

However I would not expect this paper to necessarily inspire much discussion within the ICML community, firstly because of a lack of a clear position, and secondly because TTS evaluation is likely not the central cocern of much of the community, so the releance would be to a smallish subset of it.

**Support:**

3

---

> ### Author Rebuttal · Authors · 2026-03-30
>
> Thank you for providing your valuable feedback! Please find below the responses to each comment.
>
> ---
>
> **[W1]** *"The paper does not really state a concrete, well-defined position. Instead it argues for a number of loosely related improvements..."*
>
> Thanks for your comment. We respectfully disagree. As stated in the introduction, our position is that: ***"Responsible Evaluation is essential and urgent for the next phase of TTS development."*** The concept of Responsible Evaluation is operationalized through a progressive three-level hierarchy:
>
> - *Level 1: fidelity and accuracy*
> - *Level 2: comparability, standardization, and transferability*
> - *Level 3: ethical and risk oversight*
>
> These levels are organized as a progressive structure rather than *"a number of loosely related improvements"*: without reliable metrics (Level 1), cross-system comparisons (Level 2) are not meaningful; without both, ethical oversight (Level 3) cannot be grounded. Through this framework, we critically examine current evaluation practices and identify systemic shortcomings that require urgent correction.
>
> We acknowledge that there remains room to further improve clarity; at the same time, we note that three other reviewers recognized our paper as presenting a clear position.
>
> ---
>
> **[W2]** *"most of the problems with the evaluation are relatively obvious, and the main reason they are not being addressed is that addressing them is costly and involves tradeoffs."*
>
> Thanks for your point. We agree that some problems in TTS evaluation appear *relatively obvious*. However, their persistence is precisely what makes a position paper necessary. Awareness in the TTS community remains inconsistent and inadequate. **Even in very recent top-tier publications (ICML, NeurIPS, ACL), TTS evaluation [1, 2, 3] still exhibits recurring issues**, including:
>
> - Different SOTA systems claim SOTA on different LibriSpeech subsets.
> - Comparison tables mix results from vastly different evaluation sizes (e.g., 40 vs. 1,234 samples).
> - Baseline results cited in tables are not derived from the same evaluation set as in the original work.
> - Limited reporting of subjective evaluation protocols raises concerns about possible cherry‑picking.
>
> Our empirical case studies in the Appendix further demonstrate the consequences of such inconsistencies. **Addressing them requires rigor, not cost**, and ignoring them risks obscuring genuine progress.
>
> **References**:
>
> [1] DiTAR: Diffusion Transformer Autoregressive Modeling for Speech Generation. ICML 2025
>
> [2] F5-TTS: A Fairytaler that Fakes Fluent and Faithful Speech with Flow Matching. ACL 2025
>
> [3] MaskGCT: Zero-Shot Text-to-Speech with Masked Generative Codec Transformer. NeurIPS 2025
>
> ---
>
> **[W3]** *"The recommendations in 5.5 related to ethical oversight are rather generic."*
>
> Thanks for your constructive feedback. We agree that Sec 5.5 could be more specific. In the revision, we will further elaborate by defining a set of risk dimensions and describing the key evaluation components for each. This extension aligns with our goal in Level 3: systematic assessment of misuse, forgery, privacy, and security risks, rather than leaving them as end-of-paper ethical statements.
>
> ---
>
> **[W4]** *"Figure 1 seems too handwavy to be useful in a scientific venue."*
>
> Thanks for your comment. We would like to clarify that Figure 1 is intended to provide a visual synthesis of the co-evolution between TTS technology and evaluation practices across three eras (statistical parametric, end-to-end deep learning, and foundation models), corresponding to the discussion in Sec 2. Its purpose is to motivate our central claim that evaluation has not kept pace with technological advances. The figure provides a compact overview that grounds the detailed analysis in Secs 3–5.
>
> ---
>
> **[W5]** *"I would not expect this paper to necessarily inspire much discussion within the ICML community...because TTS evaluation is likely not the central concern of much of the community..."*
>
> Thanks for your comment. We would like to clarify the **broader relevance of our work**:
> - As voice interaction becomes increasingly important in human-computer interaction, modern foundation models (e.g., GPT-4o, Gemini) are integrating speech generation as a core capability, making evaluation of speech generation systems directly relevant to the broader machine learning community.
> - Concerns such as audio deepfakes and voice cloning have escalated into mainstream machine learning challenges and are actively discussed at NeurIPS, ICML, and by policymakers worldwide.
> - The ICML Position Track explicitly calls for discussion on *"Regulation of ML technology (licensing, evaluation, disclosures, post-deployment monitoring...)"* and *"Ethical considerations when conducting ML research and deploying ML systems"*. Responsible evaluation of highly realistic TTS directly aligns with this goal and contributes to the type of forward-looking discussion the track aims to encourage.

---

> > ### Author Rebuttal · Reviewer_Vzdz · 2026-04-02
> >
> > The main weakness is in my view not resolved, and not easy to resolve without major changes to the paper:
> >
> > [W1] I think we largely have to agree to disagree on this. The title of the paper doesn't state a position at all. The introduction is more explicit, but the position "Responsible Evaluation is essential and urgent for the next phase of TTS development" is overly broad and lacks focus, in my interpretation of what the position paper call solicits.
> >
> > The other weaknesses I mentioned are largely resolved, or minor issues.
> >
> > [W2] The authors highlight the cases where the issue is not the cost but lack of rigor -- point taken.
> >
> > [W3] The authors promise to add more specifics on ethical oversight.
> >
> > [W4] The authors calrify the point of the figure. I'm not convinced, but either way, this is a minor issue.
> >
> > [W5] The authors spell out broader relevance of the issues with evaluation of TTS systems.
> >
> > I am therefore moderately increasing my score.

---

### Official Review · Reviewer_fchT · 2026-03-13

**Significance:** 4
**Argument Clarity:** 4
**Rating:** 5
**Confidence:** 5

**Questions:**

1. For Level 1, while you rightly point out the ceiling effect of Mean Opinion Score (MOS), what specific alternative human-evaluation protocols do you believe hold the most promise for high-performing systems?

2. Regarding the development of transferable metrics (Level 2), what specific steps or incentive structures can the community implement to establish universally accessible baseline datasets that do not rely on hard-to-access reference speech?

3. How do you propose the community practically standardizes adversarial risk evaluations (Level 3) given the rapid, continuous evolution of deepfake construction and attack vectors?

**Alternative Views Section:**

Yes

**Compliance With Llm Reviewing Policy A Conservative:**

Affirmed.

**Discussion Potential:**

4

**Paper Summary:**

The paper advocates for the urgent adoption of a "Responsible Evaluation" framework for Text-to-Speech (TTS) systems, arguing that current practices fail to capture the societal impacts and true limitations of modern models. The authors structure their position across three progressive levels:

Level 1: Fidelity and Accuracy calls for metrics that genuinely reflect perceptual quality and model capabilities, critiquing the current over-reliance on flawed objective and subjective metrics.

Level 2: Comparability, Standardization, and Transferability argues for scientific rigor and transparency to ensure fair cross-system comparisons, highlighting current inconsistencies in dataset usage and reporting.

Level 3: Ethical and Risk Oversight emphasizes the need to evaluate TTS systems for bias, misuse potential, data provenance, and forgery risks.

The paper's core contribution is a systematic diagnosis of current evaluation shortcomings paired with actionable recommendations for each of the three levels.

**Position:**

Yes

**Position In Title:**

Yes

**Related Work:**

4

**Strengths And Weaknesses:**

Strengths:
Relevance: The position is highly relevant given the recent proliferation of zero-shot TTS and the associated rise in audio deepfakes and telecom fraud. The call for ethical evaluation is urgently needed.

Strong Evidence: The critique of current metrics is robust and supported by concrete empirical case studies. For instance, the authors demonstrate how Word Error Rate (WER) becomes perceptually insensitive at low error rates , how varying LibriSpeech subset sizes drastically alter results , and how SIM-o computation protocols differ.

Weaknesses:
Implementation Gap: The paper could benefit from a deeper exploration of how the proposed Level 3 ethical evaluations can be operationalized. While the authors recommend establishing standardized protocols for testing against impersonation, they do not suggest specific mechanisms or datasets to achieve this.

Abstract Recommendations: Some Level 1 recommendations, such as advocating for "greater attention to perceptual validity", remain somewhat abstract. Proposing specific algorithmic or statistical adjustments to current models could strengthen this section.

**Support:**

4

---

> ### Author Rebuttal · Authors · 2026-03-30
>
> We sincerely thank you for providing your constructive and positive feedback! Please find below the responses to each comment.
>
> ---
>
> **[W1]** *"The paper could benefit from a deeper exploration of how the proposed Level 3 ethical evaluations can be operationalized. While the authors recommend establishing standardized protocols for testing against impersonation, they do not suggest specific mechanisms or datasets to achieve this."*
>
> Thank you for this important suggestion. In the revision (with an extra page), we will further operationalize Level 3 by introducing a set of standardized risk dimensions, including targeted impersonation, biometric/ASV bypass, fraud-oriented deception, and voice assistant command injection. For each dimension, we will describe the concrete evaluation components, such as prompt conditions, attack objectives, success rubrics, and reporting formats. This extension aligns with our goal in Level 3: systematic assessment of misuse, forgery, privacy, and security risks, rather than leaving them merely as end-of-paper ethical statements.
>
> ---
>
> **[R1]** *"Some Level 1 recommendations, such as advocating for 'greater attention to perceptual validity', remain somewhat abstract. Proposing specific algorithmic or statistical adjustments to current models could strengthen this section."*
>
> Thanks for your suggestion. In the revision, we will make the notion of *greater attention to perceptual validity* more concrete by grounding it in evaluation protocols. Specifically, we will clarify that for high-performing systems, comparative human evaluation protocols (e.g., pairwise preference, CMOS, and MUSHRA-style designs) should be prioritized. We will also emphasize that predicted MOS should be reported together with uncertainty or confidence intervals, to avoid over-interpreting small score differences.
>
> ---
>
> **[Q1]** *"For Level 1, while you rightly point out the ceiling effect of Mean Opinion Score (MOS), what specific alternative human-evaluation protocols do you believe hold the most promise for high-performing systems?"*
>
> We recommend that, in such cases, a practical and reliable approach is to combine:
> - comparative evaluation protocols (e.g., pairwise preference, CMOS, or MUSHRA-style multi-system comparisons) applied to diverse, expressive, and textually complex evaluation sets (e.g., CV3-Eval, EmergentTTS-Eval).
> - supplementary dimension-specific assessments (e.g., controllable, dialect, cross-lingual, multilingual, long-form).
>
> ---
>
> **[Q2]** *"What specific steps or incentive structures can the community implement to establish universally accessible baseline datasets that do not rely on hard-to-access reference speech?"*
>
> **The inaccessibility of reference speech mainly stems from benchmark under-specification**: VALL-E [1] evaluated on a subset of LibriSpeech *test-clean* but did not release the prompt list, only describing its construction procedure. This triggered fragmentation, where many subsequent works redefined their own subset and prompt list, making cross-paper comparisons infeasible even when the same corpus was used. Subsequent benchmarks, such as Seed-TTS eval [2], demonstrate that explicitly releasing the prompt list effectively resolves this issue.
>
> We therefore propose advancing on both technical and community-mechanism fronts:
> - On the technical side, we advocate building openly licensed, publicly available benchmarks with unified prompt lists, task definitions, dataset splits, and reporting templates.
> - On the community side, we advocate establishing a centralized leaderboard (analogous to the Open ASR Leaderboard [3]), which aggregates multiple benchmarks and maintains baseline records. This would incentivize researchers to evaluate against a shared reference point.
>
> **Reference**:
>
> [1] Neural Codec Language Models are Zero-Shot Text to Speech Synthesizers
>
> [2] Seed-TTS: A Family of High-Quality Versatile Speech Generation Models
>
> [3] Open ASR Leaderboard: Towards Reproducible and Transparent Multilingual and Long-Form Speech Recognition Evaluation
>
> ---
>
> **[Q3]** *"How do you propose the community practically standardize adversarial risk evaluations (Level 3) given the rapid, continuous evolution of deepfake construction and attack vectors?"*
>
> We agree that adversarial risks evolve rapidly and cannot be fully captured by a static attack set. We advocate standardizing the evaluation framework and reporting protocols, rather than fixing specific attack instances. Concretely, we suggest a hybrid design: a fixed core benchmark covering high-risk scenarios (e.g., impersonation, ASV bypass), combined with a rolling auxiliary suite to track emerging threats, ensuring adaptability while remaining cross-paper comparability.

---

> > ### Author Rebuttal · Reviewer_fchT · 2026-04-04
> >
> > Thank you authors for your detailed responses, all my questions have been addressed, and I stand by my scores.

---

### Decision · Program_Chairs · 2026-04-30

**Decision:**

Accept (regular)

**Comment:**

The paper presents an important position, and all the reviewers are positive about it. I, too, agree that TTS/ASR evaluation has to be done more responsibly in the current ML landscape. I vote that the paper be delivered in the oral presentation mode.